# Impact on Visual Acuity in Neovascular Age Related Macular Degeneration (nAMD) in Europe Due to COVID-19 Pandemic Lockdown

**DOI:** 10.3390/jcm10153281

**Published:** 2021-07-25

**Authors:** Carolina Arruabarrena, Mario Damiano Toro, Mehmet Onen, Boris E Malyugin, Robert Rejdak, Danielle Tognetto, Sandrine Zweifel, Rosa Giglio, Miguel A Teus

**Affiliations:** 1Department of Ophthalmology, University Hospital of Alcalá de Henares, Universidad de Alcalá, 28802 Alcalá de Henares, Spain; miguelteus@gmail.com; 2Department of Ophthalmology, University Hospital of Zurich, University of Zurich, 8091 Zurich, Switzerland; toro.mario@email.it (M.D.T.); sandrine.zweifel@usz.ch (S.Z.); 3Faculty of Medical Sciences, Collegium Medicum, Cardinal Stefan Wyszyński University, 01815 Warsaw, Poland; 4Chair and Department of General and Pediatric Ophthalmology, Medical University of Lublin, 20079 Lublin, Poland; robertrejdak@yahoo.com; 5Department of Ophthalmology, Ankara City Hospital, Ankara 06800, Turkey; mehmetonenster@gmail.com; 6S. Fyodorov Eye Microsurgery Federal State Institution, Russian Federation, 127486 Moscow, Russia; boris.malyugin@gmail.com; 7Eye Clinic, Department of Medicine, Surgery and Health Sciences, University of Trieste, 34134 Trieste, Italy; tognetto@units.it (D.T.); giglio.rosam@gmail.com (R.G.)

**Keywords:** COVID-19, AMD, SARS-CoV-2, neovascular age related macular degeneration

## Abstract

This is a retrospective, multicenter study of consecutive patients with nAMD scheduled for a visit and/or a treatment with an intravitreal injection (IVI) during the 3 months before lockdown in the Ophthalmology Departments of six centers of Europe.The study was conducted on 546 patients, of which 55.13% were females, almost 100% of the patients were White/Caucasian race, and 71.53% of the patients presented a type 1 macular neovascularization (NVM). A total of 62.82% of patients (343 patients) that were on scheduled clinic visits and/or intravitreal injection treatment during the 3 months before the quarantine did not attend either to visit or for treatment during the lockdown. The mean number of injections during the lockdown was significantly reduced. This was followed by a significant reduction in the mean best-corrected visual acuity (BCVA) between the 3 months before the lockdown (mean BCVA of 60.68 ± 19.77 letters) and 6 months after lockdown (mean BCVA of 56.98 ± 22.59 letters). Patients with better BCVA before the lockdown and the ones showing neovascular activity were more likely to attend their scheduled visits and/or IVI treatments. The COVID-19 pandemic and the lockdown have led to a decrease in the number of IVI treatments in patients with nAMD, evidencing a significant vision loss at 6 months.

## 1. Introduction

The coronavirus disease 2019 (COVID-19) was first reported in early December 2019 in Wuhan city, China and, subsequently, the disease spread through the country in approximately 3 months. In addition to the strategy of early identification, contact tracing, isolation, and treatment, China locked down several major cities [1]. However, major outbreaks were reported in many other countries, including the United States, Italy, Spain, Iran, South Korea, and Japan, so the World Health Organization (WHO) declared COVID-19 a pandemic on 11 March 2020 [2]. Furthermore, on 14 April 2020, it was confirmed that up to 210 countries and territories were affected, with more than 1,900,000 SARS-CoV-2 cases worldwide and more than 110,000 deaths [1]. By March 2021, there were 118.5 M cases confirmed COVID-19 cases and 2.64 M deaths worldwide. (https://ourworldindata.org, accessed on 14 March 2020)

Some countries considered treating COVID-19 as an influenza virus infection, without actively identifying the infected people, or implementing isolation or containment. Most European countries tried to prevent major outbreaks and flatten the peak of incidence. The duration of the first lockdown reported by each participating site of the study differed, depending on the pandemic situation in their country ranging, from 47 days in Switzerland, to 76 days in Russia, 81 days in Spain, up to 84 days in Poland and Turkey.

The COVID-19 pandemic has drastically modified how outpatient care is given in healthcare practices all around the world. Thus, during the worst phases of the COVID-19 outbreak, in order to reduce the chance of transmitting the virus to patients and healthcare personnel, providers have deferred elective and preventive visits, and outpatient visits have been restricted to more urgent and emergent care [3,4,5,6,7,8].

Ophthalmologists are in a high-risk category for COVID-19 infection due to close proximity with patients during their examination (conjunctival, tear secretions, and aerosol secretions) [4]. Thus, clinical practice of high-volume specialties not directly involved in COVID-19 care, such as ophthalmology, were particularly affected. Non-urgent surgeries and consultations were canceled or postponed, and in some cases, ophthalmologists were deployed in general medicine departments or in intensive care units [3,9,10,11]. In this critical period, several organizations and international ophthalmology societies produced general guidelines for patients’ management during the pandemic; the safety measures for handling elective and urgent surgeries and appointments to minimize the risk of infections for patients and physicians [12,13,14,15,16] was agreed upon. They also provided some recommendations in the prioritization process of patients with an increased risk of vision loss if left untreated [12,13,14,15,16].

Age-related macular degeneration (AMD) is the leading cause of visual disability among patients over 60 years old [17], a vulnerable age range with frequent comorbidities considered at high-risk of mortality caused by COVID-19 infection. The “dry” (atrophic) form is the most prevalent and advanced form of AMD. It is characterized by a slow, progressive dysfunction of the retinal pigment epithelium, photoreceptor loss and retinal degeneration [18]. On the contrary, the “wet” (neovascular) form, even if less frequent, is responsible for 90% of acute blindness due to AMD [18]. Neovascular AMD (nAMD) is associated with a vascular endothelial growth factor (VEGF) increase and blood vessels growing de novo; therefore, the intravitreal injection (IVI) of anti-VEGF agents is an effective treatment to inhibit neovascularization and prevent a gradual vision loss. To date, anti-VEGF agents are widely used as a pillar of nAMD treatment, to halt nAMD progression and provide over a 90% chance of stabilizing or increasing central vision after two years of treatment [19].

However, patients often require multiple IVIs and it has been shown that any delay or suspension in essential eye procedures may cause significant and rapid vision impairment that may be irreversible. Indeed, anti-VEGF treatment efficacy is time-dependent, and it correlates strongly with visual acuity at the time of the first injection [20], the degree of compliance to the applied regimen and the number of IVIs during the maintenance period [21,22].

The higher rates of cancellations and missed outpatient appointments as a response to COVID-19 significantly reduced both the in-person visits and IVI volumes during the COVID-19 quarantine [23,24,25,26].

To our knowledge, to date, there are only a few studies analyzing the early impact of the COVID-19 outbreak on BCVA changes, due to the delay or block in retinal diseases treatments. Additionally, these studies have potential bias and limitations, such as a heterogeneous study cohort or a limited number of patients enrolled, and the different protocols adopted in countries and periods of lockdown [27,28,29].

This study aimed to estimate the impact of the lockdown and other security measures implemented due to COVID-19 on patients with nAMD in different European Ophthalmology Departments. As the COVID-19 pandemic has reached a scale that total eradication is unlikely and the recurrence of outbreaks in the future is possible, our findings might be helpful in the management of nAMD patients during future waves of the COVID-19 outbreak.

## 2. Material and Methods

### 2.1. Study Design

A retrospective, international, multicenter, observational study in patients with nAMD under routine clinical care in the Ophthalmology Departments of six centers distributed among different parts of Europe was conducted. The centers involved were: Fyodorov Eye Microsurgery Federal State Institution, Moscow, Russian Federation; Ankara City Hospital, Ankara, Turkey; Medical University of Lublin, Lublin, Poland; University Hospital “Príncipe de Asturias,” Universidad de Alcalá, Alcalá de Henares, Madrid, Spain; University of Trieste, Trieste, Italy; University Hospital of Zurich, Zurich, Switzerland.

This multicenter retrospective study was performed receiving approval from the Institutional Review Board (IRB) of each site. Written informed consent was not required due to the retrospective nature of the study. All study related procedures were performed in accordance with good clinical practice (International Conference on Harmonization of Technical Requirements of Pharmaceuticals for Human Use [ICH] E6), the Declaration of Helsinki and applicable EMA regulations.

The primary aim of this study was to measure the impact of the adoption of the precautionary measures developed to address the COVID-19 pandemic on the medical activity volumes and the visual outcomes of patients with nAMD.

### 2.2. Data Collection

A comprehensive medical record review of all consecutive nAMD patients scheduled for a visit and/or a treatment with an IVI during the 3 months before lockdown in each country was performed.

In the current study, patients were included if they were ≥50 years old and had a diagnosis of nAMD. In case of bilateral involvement, only one eye of each patient (right eye) was selected and used for the analysis to decrease the risk of any bias.

Baseline demographics (age, gender, type of macular neovascularization (NVM) [30]) were collected. At each visit, BCVA measurements using Early Treatment Diabetic Retinopathy Study (ETDRS) charts were performed, although in some centers Snellen BCVA was collected and then converted to Early Treatment Diabetic Retinopathy Study (ETDRS) letters [31]. The number of visits, the number of injections and the change in mean visual acuity in 4 periods were also collected at 4 time points: 6 and 3 months before the lockdown, during the lockdown, and 6 months after it. The absenteeism of patients at the originally planned visits or IVI during the study period and the associated- demographics were also measured.

### 2.3. Statistical Analysis

Categorical variables were described using frequencies, percentages, and unadjusted odds ratios. Continuous variables were described using either means with standard errors or medians with interquartile range. Relationships between categorical variables were assessed using Fisher’s exact test and Chi-Square test, while relationships between continuous variables were assessed using independent samples *t*-test and ANOVA test. The significance level of *p* <0.05 was used for primary and secondary outcomes. Analyses were performed using Statview SE+ Graphics software (Abacus Concept Inc., Berkeley, CA, USA).

## 3. Results

### 3.1. Demographic Characteristics

The number of patients included (population size) in the study was 546. The mean age of the patients was 79.36 ± 8.83 years (range, from 51 to 97 years) and 55.13% were females. Almost 100% of the patients were of White/Caucasian race (only 1 patient was not Caucasian) and 71.53% of the patients presented a type 1 NVM.

The mean BCVA 6 months before lockdown was 60.71 ± 19.34 letters. The mean follow-up of these patients was 3.25 ± 2.29 years (range, from 6 months before the lockdown to 11 years). A total of 6.67% patients had bilateral disease and 26.23% had only one functional eye (BCVA ≤ 35 letters). Lockdown duration was different in the different countries studied and ranged from 1.5 months (47 days) to 2.75 months (84 days) with a mean of 2.49 ± 0.45 months.

There were some statistically significant differences in the demographic characteristics of the patients and the characteristics of the disease among the European countries. Table 1 and Figure 1 show the demographic characteristics of the population and the differences between the countries. The most important differences were the mean age of the patients, that were significantly younger in Russia, Turkey and Poland with a mean age of 69.55 ± 12.69, 73.13 ± 9.23 and 75.94 ± 8.40 years, respectively. The mean BCVA was also higher in these three countries. The type of NVM case mix was worst in Switzerland, Spain and Italy with more NVM type 2 and type 3, and the patients from Switzerland had longer follow up since diagnosis.

A total of 62.82% of patients (343 patients) that were on routine care visits and/or intravitreal injection treatment during the 3 months before the quarantine, discontinued the follow-up care, either because they did not attend the scheduled visits or treatment, or because the appointments were re-scheduled by the clinic in order to prioritize other cases. Moreover, 12.45% of the patients did not return to visits/treatment even during the 6 months after the lockdown, and 1.46% (8 patients) died due to COVID-19 disease.

### 3.2. Data Analysis

The number of injections during the lockdown was significantly reduced from a mean of 1.18 ± 0.78 injections per month during the 6 months before the lockdown to a mean of 0.18 ± 0.27 injections per month during the lockdown (*p* = 0.0001). This statistically significant reduction in the number of injections was followed by a significant reduction in the mean BCVA in the eyes of patients that attended the clinics during the 6 month post-lockdown period as compared to the 3 months before the lockdown values (Table 2), and also compared with the 6 months before lockdown values (−3.43 ± 13.30 letters and −3.68 ± 13.7 letters loss respectively), *p* = 0.0001 for both comparisons.

The mean number of injections per month during the lockdown was significantly different among the different countries; while Turkey has the lower 0.004 ± 0.04 injections per month, Switzerland, Poland and Russia injected the higher, 0.39 ± 0.20, 0.28 ± 0.26 and 0.22 ± 0.24 injections per month, respectively. The mean number of visits per month during the lockdown was similar among the different countries, with the lowest value seen in Turkey and the highest ones in Russia and Poland.

There were no statistical association between the discontinuation of the follow-up during the lockdown and the age, sex, duration of the lockdown and the presence of only one functional eye (Table 3). However, we found statistically significant association between the activity and the BCVA before the lockdown. Both patients with the best BCVA before the lockdown and the more active ones continued their scheduled visits and/or IVI treatments (Table 3).

The mean change in BCVA after lockdown comparing patients that continued the follow-up during the lockdown and those who discontinued it was not statistically significant; however, it is important to remark that number of injections and visits 6 months after the lockdown was significantly higher in the group that maintained their scheduled visits and/or IVI treatment during the lockdown.

## 4. Discussion

Interestingly, the demographic characteristics and types of the disease of patients with nAMD were quite different between different countries in Europe. Indeed, patients from Switzerland, Spain, and Italy were older and had a more complex combination of NVM type cases with more type 2 and type 3 NVM cases. Long-term (over 4 years) visual acuity results in patients with type 2 and mixed-type lesions have been shown to be lower compared to type 1 NVM’s [32]. In addition, risk of macular atrophy development has been reported to be increased in the presence of type 3 NVM.

On the other hand, mean bilateral disease and one only functional eye were similar between countries. Patients were statistically significantly younger in Russia, Turkey, and Poland and had a better BCVA in Poland and Turkey. The mean age was 79.36 ± 8.83 years, and there were more women than men affected, as usual in nAMD [33].

Switzerland was the country with the longest follow-up of patients and had the highest number of visits per patient during the 6 months before the lockdown, followed by Poland and Russia. The average number of visits during the 6 months before the confinement was 3.13 ± 1.45, a bi-monthly visit, similar to the results reported by Holz et al., in the AURA Study [34] carried out in 2016, and slightly lower than the most recent LUMINOUS study [35], which found a mean of 8.8 visits in a 12-month follow-up.

The mean number of injections before the lockdown was 2.65 ± 1.51 during the 6-month follow-up, according to the mean number of injections revealed in the AURA and LUMINOUS studies [34,35], 5.4, and 5.0 injections, respectively, in a 12-month follow-up. Conversely, Poland and Turkey had a more proactive treatment regimen with significantly more injections (2.94 ± 1.25 and 2.96 ± 1.07 injections, respectively). More than half of the patients (62.7%) of the whole study population were active at the last follow-up visit before lockdown and 26.23% had only one functional eye.

We have detected (during and in the post-lockdown period) a decrease in the number of visits and in the number of injections, similar to other study groups in Europe and the USA [23,24,25,26]. Due to the very different periods of confinement between countries, we have calculated the number of visits and injections per month in order to compare the influx of patients before, during and after the lockdown.

The number of visits observed in our study decreased from a mean of 1.40 ± 0.82 visits per month during the 6 months before lockdown to 0.12 ± 0.24 visits per month during the lockdown. Even so, it remained reduced during the 6 months after the lockdown with an average of1.05 ± 0.87 visits per month, which represents roughly two-thirds of those that existed before the pandemic. Warwick et al. [36] found that the number of referrals of patients with nAMD in four large hospital groups based in England decreased on average by 72% (range 65 to 87%) in April 2020 compared to April 2019. They calculated that a 3-month treatment delay could result in more than 50% relative increase in the number of eyes with vision equal or below 0.1 and a relative decrease of 25% in the number of eyes with driving vision in one year.

It is well known that a decrease in the number of injections provokes a loss in visual acuity in patients with nAMD [17,19,20,21]. In our study, the mean number of injections decreased from 1.18 ± 0.78 per month during the 6 months before the lockdown to 0.18 ± 0.27 per month during the lockdown. Unexpectedly, the number of injections per month remained decreased during the 6 months after the lockdown during the so called “new normal time,” when the mean was 0.84 ± 0.82 injections per month.

This reduction in the scheduled visits and IVI treatment was associated with a statistically significant reduction in the mean BCVA after the lockdown. The mean BCVA during the 6 months before the lockdown overall was of 60.71 ± 19.34 letters and it decreased to a mean of 56.98 ± 22.59 letters during the 6 months after the lockdown. There are few real clinical practice studies on nAMD with a follow-up period of 12 months, similar to the one studied in our publication, which compared the results in BCVA achieved during the first lockdown period in Europe.“Fight Retinal Blindness” studies carried out in the European countries during the recent years disclosed a mean improved in BCVA after a 12-month follow-up period of 3.3 letters in France [37] and 3.2 letters in unpublished data from Spain. In addition, AURA and LUMINOUS studies [34,35] had similar results with an increase of 2.4 and 3.1 letters after one year of real clinical practice treatment of nAMD.

Yeter et al. [29] in Turkey found similar results of decrease in BCVA after the lockdown in a cohort of 106 nAMD patients that didn´t received any IVI treatment or visit during the lockdown, however, the follow-up period of the study is not clear.

Our results show that 62.82% of the patients did not attend scheduled visits or receive IVI treatment during the lockdown, with the range as follows: 98.7% in Turkey, 63.4% in Spain, 62.2% in Italy, 57.0% in Switzerland 45.0% in Russia and 36.1% in Poland. During the duration of the study, 8 patients died, and it is interesting to note that 5 of the deaths occurred in the Spanish site. It is very important to remark that 12.45% of patients discontinued their scheduled visits and IVI treatment even 6 months after the lockdown. Our study was not designed to find the cause of these follow-up and permanent treatment-dropouts. Preliminary checks have shown that potential causes of the loss of follow-up could be because patients changed their address or residence, or patients with fear of going to hospitals or lack of support from relatives to visit the clinic [25].

We analyzed the characteristics of the patients with discontinuation of the follow-up regime during the lockdown and we did not find statistically significant differences in age, sex, duration of the disease or in patients with only one functional eye. On the other hand, patients that attended the scheduled hospital visits had significant better BCVA a mean of 63.55 ± 17.68 letters (*p* = 0.009). Additionally, there were more patients with active neovascular lesions before the lockdown attending the scheduled visits than those who discontinued, 86.36% versus 52.9%, respectively (*p* = 0.0001).

There was no statistically significant difference in BCVA between patients who attended the follow-up visits/treatment than in those who discontinued the follow-up; probably because 6 months is a very short period to find differences and because patients in the group with maintained follow-up had better BCVA. However, we found differences in the number of visits and IVI treatments, that were significantly higher in the group that attended the scheduled visits/treatment (*p* = 0.0001 and *p* = 0.0001). It is interesting to note that some patients did not resume their visits after the lockdown. This fact might be due either to individual patient choice or because the patients who discontinued the follow-up might have greater difficulties in getting to their new appointments after having discontinued the active follow-up visits.

The main limitation in our study was the short follow-up period, where the long-term impact of the lockdown on patients’ visual outcomes is yet to be unraveled.

In conclusion, the COVID-19 pandemic and the lockdown have led to a decrease in the number of IVI treatments in patients with nAMD evidencing a significant vision loss at 6 months. Further studies will be necessary to better understand the vision recovery of this patient population.

## Figures and Tables

**Figure 1 jcm-10-03281-f001:**
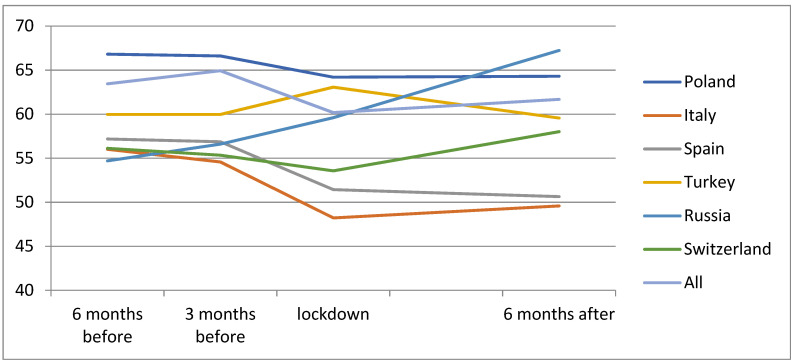
Mean evolution of the BCVA per country.

**Table 1 jcm-10-03281-t001:** Patient demographics.

Country	Poland (*n* = 94)	Italy (*n* = 105)	Spain (*n* = 144)	Turkey (*n* = 82)	Russia (*n* = 20)	Switzerland (*n* = 101)	All (*n* = 546)	*p* Value
Age (years)	75.94 ± 8.40	81.15 ± 7.02	79.43 ± 7.57	73.13 ± 9.23	69.55 ± 12.68	83.38 ± 7.43	79.36 ± 8.83	0.0001
Gender (% female)	54.26	54.29	54.86	52.44	75	62.07	55.13	ns
Bilateral disease	0%	0%	14.37%	16.33%	15.38%	17.17%	6.67%	ns
Lockdown duration	2.5	2.75	2.75	2.75	2.5	1.5	2.51 ± 0.3	
CNV type								0.001
type 1	89.36	49.52	68.06	92.68	80	49.50	71.53	
type2	10.64	46.67	18.06	7.32	20	43.56	24.37	
type3		2.86	13.19			6.93	3.83	
type 1 aneurismatic		0.95	0.69				0.27	
Disease follow-up	3.11 ± 0.12	3.07 ± 0.23	2.97 ± 0.18	2.40 ± 0.19	2.74 ± 0.75	6.91 ± 0.27	3.25 ± 0.09	0.0001
Only one functional eye	19.15	35.24	27.08	34.15	15	18.00	26.23	ns
AV > 70	50.00%	26.67%	40.28%	36.59%	45.00%	59.407%	41.54%	
AV > 50	98.94%	69.52%	73.61%	80.49%	70.00%	82.17%	79.30%	
AV < 30	1.06%	8.57%	13.89%	7.32%	15.00%	11.88%	8.97%	
Mean BCVA 6 months before the lockdown	66.81 ± 10.54	56.01 ± 18.12	57.19 ± 23.14	59.97 ± 17.94	54.7 ± 25.93	66.36 ± 19.00	60.71 ± 19.34	0.003
Mean BCVA 3 months before the lockdown	66.60 ± 11.56	54.57 ± 18.74	56.87 ± 24.09	62.13 ± 18.18	56.6 ± 26.38	67.13 ± 17.91	60.68 ± 19.77	0.0001
Number of visits 6 months before the lock down	3.85 ± 0.64	2.08 ± 1.02	2.67 ± 0.88	3.02 ± 1.00	6.05 ± 3.50	3.67 ± 1.35	3.13 ± 1.45	0.0001
Number of IVI 6 months before the lockdown	2.94 ± 1.24	2.81 ± 1.48	2.20 ± 1.60	2.96 ± 1.070	1.9 ± 1.11	2.71 ± 1.83	2.65 ± 1.51	0.0001
% of active CNV before the lock down	91.49%	76.19%	44.44%	28.05%	95.00%	77.0%	65.17%	0.0001
% patients that discontinued the follow-up during the lockdown	36.1%	63.4%	62.2%	98.7%	45.0%	57.0%	62.8%	0.0001

**Table 2 jcm-10-03281-t002:** BCVA and number of visits and IVIs in each period.

	6 Months before the Lockdown (*n* = 526)	During the Lockdown	6 Month after the Lockdown	*p* Value
		(*n* = 143)	(*n* = 469)	
Mean BCVA	60.71 ± 19.34	59.18 ± 19.32	56.98 ± 22.59	*p* = 0.001
Number of IVIs	2.65 ± 1.51	0.38 ± 0.57	1.84 ± 1.52	*p* = 0.001
Number of IVIs (per month)	1.18 ± 0.78	0.18 ± 0.27	0.84 ± 0.82	
Number of visits	3.13 ± 1.45	0.33 ± 0.63	2.24 ± 1.46	*p* = 0.001
Number of visits (per month)	1.40 ± 0.82	0.12 ± 0.24	1.05 ± 0.87	

**Table 3 jcm-10-03281-t003:** Differences between patients that continued on schedule visits and/or IVI treatment and patients that discontinued visits and/or treatment during the lockdown.

All Patients *n* = 546	Discontinued Follow-Up *n* = 343	Continued Follow-Up *n* = 203	*p* Value
Age (years)	79.93 ± 8.8	78.4 ± 8.7	0.053
Gender (%female)	53.0%	38.8%	0.3
Only one functional eye	25.15%	27.72%	0.5
Duration of the disease (years)	3.18 ± 2.22	3.41 ± 2.43	0.2
Mean BCVA 6 months before lockdown	59.02 ± 20.08	63.55 ± 17.68	0.009
AV ≥ 70 letters	38.7%	52.6%	0.02
AV ≥ 65 letters	51.7%	63.5%	0.009
% of eyes with activity of the lesion before lockdown	52.9%	86.36%	0.0001
Lockdown duration (months)	2.45 ± 0.43	2.42 ± 0.49	0.4
Mean change in BCVA 6 months after lockdown (letters)	−3.41 ± 12.80	−3.45 ± 14.04	0.9
Nº of visits during the 6 months after lockdown	2.02 ± 1.28	2.61 ± 1.66	0.0001
Nº of injections during 6 months after lockdown	1.50 ± 1.51	2.36 ± 1.39	0.0001

## Data Availability

The data presented in this study are available on request from the corresponding author. The data are not publicly available due to ethic issues.

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
