# Peer review of "Impact on Visual Acuity in Neovascular Age Related Macular Degeneration (nAMD) in Europe Due to COVID-19 Pandemic Lockdown"

_jcm, 2021, doi:10.3390/jcm10153281_

Round 1

Reviewer 1 Report

COVID-19 pandemic, caused by the spread of SARS-CoV-2 virus, has fundamentally changed all aspects of human society worldwide. Especially, the enforced lockdown policy impacts the clinic follow-up and the intervention of ophthalmological medicine. In this manuscript, the authors performed a multicenter, retrospective study with 546 wet AMD patients from six centers of Europe. They found that lockdown policy significantly reduced the clinic visits or intravitreal injection treatment, leading to the BCVA loss without statistics significances. Overall, the whole study got similar conclusions from prior studies, as well as the heterogeneous study cohort or other limitations from previous literature. This study suggests the requirement of improving the management of wet AMD patients during COVID-19 pandemic. I have few suggestions:

1. In the abstract, the term "BCNA" should be defined in its first appearance.

2. In line 41 of the introduction, COV-Sars2 should be SARS-COV-2

3. In the result, the citation of table 3 is missing

4, In previous studies, the significant BCVA impairment was only found in nAMD patients with diabetic complications, not the other nAMD patients. Is there any information on the patients used in this study about the presence of diabetic complications, or other conditions? This should be discussed.

Author Response

Response to Reviewer 1:

1- In the abstract BCVA has been defined as best-corrected visual acuity.

2- In line 41 COV-Sars2 has been changed to SARS-COV-2.

3- In the results section we now cite table 3.

4- Unfortunately, we did not collect information about the presence of diabetic complications or other conditions in the patients studied.

Reviewer 2 Report

Thank you for this interesting submission. It is an important study to show the hard task ahead to get ophthalmology services back on track after the ravages of the pandemic. A few points for consideration:

In the introductory paragraphs, it might be also useful to note that some ophthalmologists were redeployed to the 'front line' in some European countries to treat COVID with other medics, which then had a knock-on impact on ophthalmology services.

You reference Warwick et al - the UK had quite a different COVID experience compared to other European countries, and a stretched NHS service, so results are likely to be different. It would have been useful to include data from the UK; but maybe a follow up study could do this.

You introduce the acronym for best corrected visual acuity in line 77 without definition; please can you include it in full the first time it is introduced.

Line 212 - sp error / grammar error with 'rage'.

General formatting - please can you review the entire paper to ensure that the spacing between words is appropriate.

Author Response

Response to Reviewer 2:

1- We have tried to summarize the situation of ophthalmologists during the COVID pandemic in the introduction with this paragraph:

"Thus, clinical practice of high-volume specialties but not directly involved in Covid 19 care, such as ophthalmology, were particularly affected. Non-urgent surgeries and consultations were canceled or postponed, and in some cases, Ophthalmologists were deployed in general medicine departments or in intensive care units."

2- Each European country analyzed had a different experience during the COVID pandemic but with quite similar results. It will be interesting, in future studies, to include more countries, such as the UK.

3- We agree, and the acronym BCVA has been explained in the abstract, following the reviewer's suggestion.

4- The spelling error in line 212 has been changed from “rage” to “range.”

5- We have reviewed the entire paper to correct the spacing between words.